# Association between mink coronavirus (MCoV), *Campylobacter* spp., and diarrhea in farmed mink *(Neogale vison)*

Michelle Lauge Quaade[1]*, Mikael Leijon[2], Mikhayil Hakhverdyan[2], Thomas Bruun Rasmussen[3], Charlotte Kristiane Hjulsager[3], Lars Andresen[1], Karin Mundbjerg[1], Anne Sofie Vedsted Hammer[1]

**1** Department of Veterinary and Animal Sciences, University of Copenhagen, Frederiksberg C, Denmark, **2** Department of Microbiology, Swedish Veterinary Agency, Uppsala, Sweden, **3** Department of Virology and Microbiological Preparedness, Statens Serum Institut, Copenhagen S, Denmark

* mlq@sund.ku.dk

## Abstract

Diarrhea outbreaks significantly affect the health and growth of farmed mink, posing economic and welfare challenges. While various pathogens have been linked to mink diarrhea, the causes during the weaning period remain unclear. Sporadic associations of mink coronavirus (MCoV), *Campylobacter*, and gastrointestinal disease in mink have been suggested. This study investigates the occurrence and levels of MCoV and *Campylobacter* in fecal samples from Danish farm mink *(Neogale vison)* and their potential association with post-weaning anorexia diarrhea syndrome (PADS), growth-period diarrhea (GPD), or pre-weaning diarrhea (PWD). The focus of the study is on PADS, studied through two case-control setups where case farms had known problems with PADS and control farms without such problems. Animals were also diagnosed based on necropsy pathological findings. Seventeen MCoV genomes were determined from five Danish farms. This data facilitated the development of an MCoV RT-qPCR and was applied across three study groups. Results showed high prevalence of both MCoV and Campylobacter spp. in all groups. No link was found between MCoV presence or levels and PADS diagnosis on the individual level; however, farm-level analysis revealed that MCoV was 2.35 times more likely (95% CI 1.027; 5.056) on PADS case farms than controls. Mink with PWD exhibited higher MCoV levels compared to GPD cases, suggesting a role at this developmental stage. Phylogenetic analysis revealed diverse and farm-specific MCoV strains, with sequences from healthy controls forming a distinct subclade, hinting at strain-specific pathogenicity. *Campylobacter* spp. presence was not significantly associated with PADS, but higher levels were observed in mink with PADS versus those without (not tested for PWD). These findings highlight the importance of surveillance and rigorous hygiene practices on mink farms to address risks from MCoV and *Campylobacter* spp., calling for further research to clarify their roles in PADS and overall mink health.

**Data availability statement:** Data from this study is available at the Open Science Framework (OSF) under study ID m4ce2 and can be accessed via DOI: 10.17605/OSF.IO/M4CE2. All MCoV sequence data has been deposited at NCBI with accessions: PQ451274.1-PQ451290.1.

**Funding:** The author(s) received no specific funding for this work.

**Competing interests:** The authors have declared that no competing interests exist.

## Introduction

Outbreaks of diarrhea on mink farms during the pre-weaning and growth periods are a major cause of disease, reduced growth, and increased mortality in mink kits [1,2] resulting in both reduced animal welfare and economic losses. However, limited information is available on pathogens involved in mink diarrhea, and few studies have systematically evaluated the etiology of diarrhea in mink over the past decades [3–5]. While both coronaviruses (CoVs) and *Campylobacter* spp. have been suggested as causal factors of diarrhea and wasting in farmed mink, this is to our knowledge, the first report of a systematic screening for mink coronavirus (MCoV) and *Campylobacter* spp. in fecal samples from Danish farmed mink in relation to diarrhea. Previous studies have detected coronavirus particles by electron microscopy and PCR in farmed mink suffering from diarrhea in the post-weaning period [6,7], and serological studies indicated a high prevalence of coronavirus-specific antibodies on Danish mink farms [8]. Both MCoV and *Campylobacter* spp. have been isolated from clinically healthy mink kits and kits with diarrhea [9–11], leaving their roles in diarrhea syndromes unclear. It remains unknown whether the presence, quantity, or strain variation of MCoV, as well as potential infection or coinfection with *Campylobacter* spp., is associated with disease expression or mortality in the farm mink.

CoVs are enveloped, positive-sense, single-stranded RNA viruses and are known to cause respiratory, enteric, neurologic, or systemic disease in a vast variety of mammalian animal species, including humans [12]. CoVs are classified under the *Coronavirinae* subfamily within the *Coronaviridae* family and the order Nidovirales. This subfamily is divided into four genera: *Alphacoronavirus*, *Betacoronavirus*, *Gammacoronavirus*, and *Deltacoronavirus*, based on phylogenetic relationships and genomic structures [13]. *Alpha-* and *Betacoronaviruses* primarily infect mammals, while *Gamma-* and *Deltacoronaviruses* mostly infect birds, though some are capable of infecting mammals [14]. In light of the Coronavirus disease (COVID-19) pandemic caused by SARS-CoV-2, interest in zoonoses has surged and evidence of SARS-CoV-2 transmission between humans and mink on Danish mink farms [15] highlights the importance of investigating the prevalence of circulating CoVs in farmed mink.

*Campylobacter* spp. are Gram-negative, microaerophilic, curved, motile rods. Although the genus comprises 37 species and subspecies, most are considered non-pathogenic [16]. *Campylobacter* spp. are well known for causing diarrhea in humans [16] but in dogs and cats, studies have found no association between *Campylobacter* spp. in feces and diarrhea in dogs and cats, since similar isolation rates in healthy and diarrhetic animals have been reported [17–19]. In mink, *Campylobacter* spp. have mainly been associated with abortion and reproductive failure [20,21], although *C. jejuni* has also been associated with outbreaks of colitis in farmed mink [22]. In addition, *C. jejuni* was found to be absent in Danish mink kits during the lactating period but prevalent during the growth period [23,24]. Although these results generally indicated frequent occurrence of *C. jejuni* in mink without diarrhea it was found that samples from animals with signs of diarrhea exhibited a trend towards presence of *Campylobacter* and to be more prevalent on farms reporting outbreaks of

diarrhea [24]. However, the study was inconclusive regarding causality, since *Campylobacter* was also frequently found in samples from healthy animals.

The aim of this study was to investigate the association of MCoV and *Campylobacter* spp. and different diarrhea syndromes in farmed mink (*Neogale vison*), focusing on PADS but also considering pre-weaning diarrhea (PWD), and growth period diarrhea (GPD). To achieve this, 17 new MCoV genomes from Danish mink were characterized and a novel reverse transcription quantitative PCR (RT-qPCR) assay for detection of MCoV in mink fecal samples was developed. The samples included in this study consisted of both archived samples and samples collected in relation to previous studies on mink farms. Thus, these samples were not originally collected specifically for the current investigation but were included to gain preliminary insights into the presence and diversity of pathogens circulating on mink farms. Screening of these samples was conducted to evaluate associations with clinical symptoms, gross pathological findings, and farm management practices. Additionally, we investigated whether genomic variation among MCoV strains was linked to differences in pathogenicity.

## Materials and methods

### Study design

The results presented in this study is based on samples from three separate sub-studies: 1) a method development (MD) case-control study including one case and one control farm matched on geographic location and consulting veterinarian conducted in 2018; 2) an observational retrospective case-control study including samples from 5 case and 14 control farms conducted in 2018; 3) an exploratory case study including samples from animals submitted for diagnostics at University of Copenhagen (UCPH) from 43 farms conducted in 2017.

In the MD study and the observational case-control study, two levels of group classification were used simultaneously: 1) farm level, where animals originated from either a case or control farm, and 2) animal level, where each animal was diagnosed as having PADS or not, based on gross pathological findings observed during necropsy. In these two studies, control animals were selected among mink submitted for necropsy that did not exhibit clinical signs or pathological findings compatible with PADS, such as diarrhea, enteritis, or splenomegaly. Thus, animals with gross or histological evidence suggestive of PADS were excluded from the control group.

In addition, sequencing data from 6 fecal samples collected from two mink farms during a PADS outbreak in 2019 were obtained and included in the phylogenetic analysis.

**Animals and ethics.** All included samples in these studies were obtained from previously conducted research, where the animals were either found deceased on the farms or euthanized due to acute disease. Consequently, no animals were culled specifically for this research project, and thus, no ethical or laboratory animal permissions were required. If animals were euthanized in the previously conducted studies, the procedure was carried out by trained farm personnel on the farms, in accordance with Danish legislation [25].

**Method development case-control study (MD-study).** Fecal samples collected in July 2018 from two commercial mink farms were used in a case-control MD-study. Samples from one farm with PADS (case farm) and one without (control farm) were investigated. The case farm was selected based on the history of disease obtained from the consulting veterinarian which stated that there had been problems with diarrhea in the post-weaning period the previous years. In total, fecal samples from 30 mink kits from the case farm and 10 from the control farm were included. Skilled veterinary personnel performed routine necropsies, and documented findings. Fecal samples from all 40 animals were stored at −20 °C. This study aimed to develop methods for detection of pathogens, specifically MCoV, in fecal samples from mink.

**Observational retrospective case-control study.** The observational retrospective case-control study was conducted on fecal samples collected from April 1 to October 28, 2018, on 19 commercial mink farms. The selection of farms and animals is described in a previously conducted study that investigated associations between urinary tract disease in mink and animal and herd management factors [26]. Each farm had the same consulting veterinarian, and all agreed to

participate in the study. Based on information on disease history on the farms provided by the consulting veterinarian, five farms were defined as PADS case-farms and 14 as control farms. A farm would be defined as a PADS case farm if the consulting veterinarian stated that there had been problems with diarrhea in the post-weaning period the previous years. On average, 10 animals per farm were randomly selected for fecal sampling among all dead mink kits from July 2018, although the number of collected samples per farm ranged from one to 52 samples.

Skilled veterinary personnel associated with the project performed necropsies on the farms, and findings were recorded and included in this study. Based on the necropsy findings, each mink was diagnosed as either having PADS or not. A mink kit would be diagnosed with PADS if all the following criteria were fulfilled: 1) body condition score of 1 or 2 out of 5; 2) splenomegaly; and 3) no signs of other diseases or lesions. The necropsies were performed in a blinded manner, as the personnel did not know the disease status on the farm. A total of 208 fecal samples were collected, stored at −20 °C, and subsequently included in this study.

Additionally, management data on the farms were collected in the previously conducted study [26] and these data were also included in the present study. The data recorded included farm size (number of breeding females), color types of breeding females, weaning procedures, water data (water supply, type of water nipple, and availability of extra water supply at the nest box), feeding pattern and kit age at feed introduction, use of feed additives (dose, time of use, and frequency) and procedures for kit nets in the cages (use of short nets, procedures for changing, disinfecting, and manure removal of nets in addition to time of final removal).

**Exploratory case study.** To investigate the presence of MCoV and *Campylobacter* spp*.* in fecal samples from mink kits in previous years, an exploratory case study was conducted. Here, fecal samples (stored at −20 °C) from a previously conducted study in 2017 [27] were analyzed for the presence and levels of MCoV and *Campylobacter* spp.. Samples collected from mink kits submitted for diagnostics at UCPH in 2013 from the pre-weaning period, May to June (0–2 months old) and the growth period, July to November (3–6 months old) were included. Based on necropsy findings and age of the animals, the mink kits were diagnosed with either pre-weaning diarrhea (PWD) or growth-period diarrhea (GPD) or no diarrhea. No PADS outbreaks were identified in 2013 and therefore samples from mink with PADS were not included in this sample group.

## RT-qPCR and qPCR

All included fecal samples from the three different studies were analyzed for the presence and levels of MCoV and *Campylobacter* spp., as well as for the presence of Mink enteritis virus (MeV), as the latter is considered the most relevant pathogen for diarrhea in the post-weaning period [28].

**MCoV RT-qPCR.** *MCoV primer design*: MCoV genome sequences obtained from the fecal samples collected from the MD-study and the included samples from diagnostic samples collected during a PADS outbreak in were used to design a MCoV specific RT-qPCR assay. Briefly, alignment analysis between sequencing data from this study and previously published sequences from mink and ferret coronaviruses (NCBI nucleotide accession numbers HM245925, MF113046, HM245926, KX512809 LC215871, KM347965, FJ938054, KY566209 and FJ938055) revealed conserved regions. These regions were used as templates in a Primer–BLAST search (https://www.ncbi.nlm.nih.gov/tools/primer-blast/index.cgi), which gave the primer/probe combination displayed in Table 1.

The PCR amplicon was purchased as a gBlock gene fragment (Integrated DNA Technologies, Leuven, Belgium) with a defined copy number, which was used as a template in the assay optimization and was further used as a positive control.

Table 1. Primers and probe used for detection of Mink coronavirus (MCoV).

| | |
|---|---|
| MCoV_F | GTGTTGGGTTGAACCAGAYTTAA |
| MCoV_R | CCAAACTAACATAACGCTCAAGC |
| MCoV_P | FAM-GGACCACATGARTTTTGYTCGCAGC-BHQ1 |

Extraction of RNA from the exploratory case study samples were done as described previously [27]. When analyzing samples, the purified RNA samples were reverse transcribed using the iScript Advance cDNA kit (Bio-Rad Laboratories Inc., Hercules, CA) using 5 μl RNA in a total reaction volume of 20 μl. After the reactions were completed, the cDNA samples were diluted 10 times in nuclease-free water. Five μl was used in a 20 μl qPCR reaction containing 300 nM each of primers and probe and 10 μl of SsoAdvanced Universal Probes Supermix Mix (Bio-Rad Laboratories Inc., Hercules, CA). Reactions were run in duplicates on a Bio-Rad CFX96 thermocycler (Bio-Rad Laboratories Inc., Hercules, CA) using a two-step temperature cycle (95°C for 10 seconds and 62°C for 30 seconds) and repeated 39 times. Amplification control reactions using samples spiked with 5000 copies of gBlock were also included. Threshold of quantification (Cq) values were calculated using CFX Maestro Software v. 2.3 (Bio-Rad Laboratories Inc., Hercules, CA) with standard settings, and the mean Cq value was reported. Only samples with Cq values in both duplicates were considered positive for MCoV.

**Campylobacter spp. qPCR.** Preparation of fecal samples and DNA isolation were done as previously described [5]. The reaction mix was the same as described for MCoV detection and was set up in duplicates. The PCR was run for 40 cycles. Each cycle consisted of 95°C for 5 seconds and 60°C for 30 seconds. Primers and probe used for the qPCR reaction have been described previously [29] and are listed in Table 2. Threshold of quantification (Cq) values were calculated using CFX Maestro Software v. 2.3 (Bio-Rad Laboratories Inc., Hercules, CA) with standard settings, and the mean Cq value was reported. Only samples with Cq values in both duplicates were considered positive for *Campylobacter* spp. For positive control DNA extracted from Maldi-TOF verified strain of *Campylobacter jejuni* was used, whereas no template control samples were used as negative controls.

**Mink enteritis virus qPCR.** Detection of mink enteritis virus (MEV) infection was performed by qPCR, following the protocol used for MCoV, except that a SYBR Green based mastermix (SsoAdvanced Universal SYBR Green Supermix, Bio-Rad Laboratories Inc., Hercules, CA) was used. The primers used targeted the VP2 gene of MEV and their sequences are listed in Table 3. Cycle conditions were 95°C for 5 seconds and 62°C for 20 seconds with 40 cycles in total.

### Next-Generation Sequencing

**Sample preparation.** Seventy-five fecal samples were homogenized using the FastPrep-24 Sample Preparation System (M.P. Biomedicals, Irvine, CA) in 2 ml microtubes, each containing twenty 2.0 mm zirconia beads (BioSpec Products, Bartlesville, OK). The fecal homogenates were centrifuged at 10,000 x g for 10 min. at 4 °C. In addition, for five samples (sample 1–4 and 10, see Table 6), the supernatant was further filtered through a Millex-HPF HV 0.45 μm filter (Merck Millipore, Cork, Ireland) using a 2-ml syringe to remove large particles, bacteria, and host cells. Due to problems with filter clogging, this procedure was not generally applied.

**Table 2. Primers and probe used for *Campylobacter* spp. qPCR\*.**

| 16S-CampF | CACGTGCTACAATGGCATATACAA |
|---|---|
| 16S-CampR | CCGAACTGGGACATATTTTATAGATTT |
| 16S-CampP | FAM-AGACGCAATACCGTGAGGT-MGB |

\*Primers and probe have previously been published by de Boer et al. [29]

**Table 3. Primers used for mink enteritis virus qPCR.**

| MEV_VP2_F | AGA GCA TTG GGC TTA CCA CC |
|---|---|
| MEV_VP2_R | CCA ACC TCA GCT GGT CTC AT |

RNA extraction was performed using TRIzol LS Reagent (Invitrogen Thermo Fisher Scientific, Carlsbad, CA) in combination with RNeasy Mini Kit (QIAGEN, Hilden, Germany) and in-column DNase I digestion, as described previously [30]. The purified RNA was recovered in 40 µL of nuclease-free water and stored at −80 °C until further use.

Reverse transcription was performed using the SuperScript IV (Invitrogen, Thermo Fisher Scientific, Carlsbad, CA) with the FR26RV-6N random primer for tag-labelling of cDNA, followed by second-strand cDNA synthesis and sequence-independent single primer amplification (SISPA) using the FR20RV primer, as previously described [31], except for six samples (samples 12–17, see Table 6), which produced sufficient DNA concentration and not requiring SISPA. For these samples, reverse transcription was performed using the SuperScript IV First Strand Synthesis kit and random hexamers according to the manufacturer's instructions (Invitrogen, Thermo Fisher Scientific, Carlsbad, CA), and the second-strand cDNA synthesis was performed using the NEBNext enzyme mix according to the manufacturer's instructions (New England Biolabs, Ipswich, MA). Finally, the concentration of DNA was verified with the Qubit® fluorometer 2.0 (Thermo Fisher Scientific, Waltham, MA) and diluted to 0.2 ng/µL as per the Nextera XT DNA library preparation protocol (Illumina, San Diego, CA).

**Library preparation and sequencing.** The sequencing library was prepared following the Nextera XT DNA library preparation protocol (Illumina, San Diego, CA). The quality and quantity of the library were assessed with the Agilent 2100 Bioanalyzer (Agilent Technologies, Waldbronn, Germany). The libraries were then normalized to a concentration of 4 nM, pooled together, and prepared for sequencing on the MiSeq platform according to the manufacturer's guidelines (Illumina, San Diego, CA). The pooled library, at a final concentration of 12 pM, was sequenced as paired-end reads using the MiSeq 600 v3 Reagent Kit (Illumina, San Diego, CA).

## Bioinformatics

The Illumina 2x300 paired-end reads were quality trimmed using a Trimmomatic v 0.39 [32] with a sliding window of four nucleotides and required average quality score of 15. The trimmed reads were subjected to *de novo* assembly using SPAdes v 3.15.4 in rnaviral mode [33]. The assembled contigs were then classified using DIAMOND v 2.0.9 [34] with a database for classification created using the NCBI nr database of GenBank release 248 and the corresponding NCBI taxonomy databases. When a single contig was not obtained covering the whole genome but separate overlapping contigs could be further assembled to produce complete genomes, this was carried out using the QIAGEN CLC Genomics Workbench, version 24.0.1 (QIAGEN, Aarhus, Denmark).

The CLC genomics workbench was also used to align the coronavirus genomes and to carry out maximum likelihood phylogeny. Before calculating the phylogeny, the alignment was cropped to the maximum equivalent position at both ends where all sequences were existent. This led to sequence lengths in the range 27832 nt (NC_030292, the outgroup ferret CoV) to 28331 nt (sample 12, see Table 6). The maximum likelihood calculation used the neighbor-joining construction model with the Jukes-Cantor and WAG nucleotide and protein substitution models, respectively. The transition/transversion ration was set to 2.0, and 4 substitution rate categories were used. The gamma distribution parameter was set to 1. The bootstrap analysis utilized 100 replicates.

## Statistics

All statistical analyses and production of graphs were conducted in GraphPad Prism version 10.0 (GraphPad Software, San Diego, CA). Ordinary one-way analysis of variance (ANOVA) with multiple comparisons was used to compare RT-qPCR and qPCR Cq-values of MCoV and *Campylobacter* spp. in the different groups of animals. Associations between the recorded management factors and prevalence of PADS were analyzed as previously described by Mundbjerg et al. [26]. Measures of associations of disease (PADS) and exposure (MCoV/*Campylobacter* spp.) were analyzed by odds ratio (OR) with 95% confidence intervals by using Fischer's exact test. Statistical significance was defined at a 5% significance level ($p < 0.05$).

## Results

### Necropsy findings

In the MD-study, 15 of the 30 mink kits from the case farm were diagnosed with PADS based on necropsy recordings. No mink from the control farm ($n = 10$) were diagnosed with PADS and no other pathological findings were recorded.

In the observational case-control study, a total of 2.625 mink kits died or were euthanized throughout July on the 19 included farms. Of these, 248 mink kits (76 from case farms and 172 from control farms) were deemed too cadaverous for gross pathological examination, giving a total of 2.377 mink kits included for necropsy. A total of 273 mink kits were diagnosed with PADS based on necropsy recordings (137 from case farms and 136 from control farms). A summary of the included animals relative to the total number of animals on the case and controls farms respectively is shown in Table 4.

### Detection of MCoV, *Campylobacter* spp., and MEV

In total, 40 samples from the MD-study (30 from the case farm and 10 from the control farm) and 208 samples from the observational case-control study (58 from case farms and 150 from control farms) were analyzed for the presence and levels of MCoV and *Campylobacter* spp. by RT-qPCR and qPCR, respectively. In the exploratory case study, 107 samples from the pre-weaning period (98 cases and 9 controls) and 88 from the growth period (42 cases and 46 controls) were analyzed for the presence of MCoV and 73 samples from the growth period (35 cases and 38 controls) for the presence of *Campylobacter* spp.

All the included fecal samples were negative for MEV.

**Table 4. Summary of included animals relative to farm size and number of fecal samples.**

|  | Farm status[1] *Number of farms* | |
|---|---|---|
|  | Control 14 | Case 5 |
| *Dead kits in July* %[2] ± SD *Mean (min;max)* | 1.01% ± 0.27 ~117 (28;295) | 0.67% ± 1.64 198 (124;328) |
| *Necropsied kits* %[2] ± SD *Mean (min;max)* | 0.90% ± 0.27 104 (26;268) | 0.62% ± 1.33 ~183 (118;311) |
| *Mink kits on farms in week 27* Mean ± SD *(min;max)* | 11531 ± 8452 (4376;31421) | 29643 ± 17035 (3711;41855) |
| *Mink kits with PADS diagnosis*[3] %[2] ± SD *Mean (min;max)* | 0.08% ± 0.07 ~10 (1;39) | 0.09% ± 0.11 27 (12;52) |
| *Mink kits with PADS diagnosis among necropsied mink kits* %[4] ± SD | 7.4% ± 4.5 | ~15% ± 3.3 |
| *Fecal samples* Mean ± SD *(min;max)* | ~11 ± 12 (1;52) | 12 ± 5 (7;19) |

[1]Farmstatus based on the disease history of the farm obtained from the local consulting veterinarian.

[2]Relative to the total number of mink kits in week 27 on control- and case-farms.

[3]Diagnosis of PADS based on the gross pathological findings, where the following should be fulfilled: 1) body condition score of 1 or 2; 2) splenomegaly; and 3) no signs of other disease or lesion.

[4]Relative to number of autopsied mink kits on the case and control farms.

**Odds ratios.** In the MD-study, MCoV was detected in 37 mink kits (27 from the case farm and 10 from the control farm) and, of those, in 13 out of 15 mink kits diagnosed with PADS and in 24 out of 25 without PADS diagnosis based on necropsy findings. *Campylobacter* spp. was detected in 37 mink kits (29 from the case farm and 8 from the control farm), and of those, in all 15 mink kits with PADS diagnosis and in 22 out of 25 without PADS diagnosis based on necropsy findings.

In the observational case-control study, MCoV was detected in 159 mink kits (50 from case farms and 109 from the control farms) and, of those, in 28 out of 36 mink kits with PADS diagnosis and in 131 out of 172 without PADS diagnosis based on necropsy findings. *Campylobacter* spp. was detected in 177 mink kits (49 from case farms and 128 from control farms) and, of those, 32 out of 36 mink kits diagnosed with PADS and in 145 out of 173 without PADS diagnosis based on necropsy findings.

The odds ratios (OR) with an associated 95% CI for detection of MCoV or *Campylobacter* spp. in mink kits from either a PADS case farm or with a diagnosis of PADS compared to mink kits from a control farm or without a PADS diagnosis are shown in Table 5. Detection of MCoV was significantly associated with farm status in the observational case-control study, where the odds of MCoV detection in fecal samples was 2.35 (95% CI 1.027;5.056) times higher in mink kits originating from a PADS case farm compared to mink from a control farm.

In the exploratory case study, MCoV was detected in 78 fecal samples from mink kits in total (43 with PWD, 12 with GPD, and in 23 kits without GPD). MCoV was not detected in 117 samples (55 with PWD and 30 with GPD). *Campylobacter* spp. was detected in 26 mink kits (12 with GPD) and was not detected in 47 mink kits (23 with GPD).

**Levels of MCoV and Campylobacter spp.** In the MD-study, among the 37 samples where MCoV was detected, significantly higher levels of MCoV were found in fecal samples from the control farm compared to the case farm. Among the 37 samples where *Campylobacter* spp. were detected, significantly higher levels of *Campylobacter* spp. were found both in samples from the case farm and from mink kits diagnosed with PADS at necropsy compared to samples from the control farm and mink without PADS diagnosis.

In the case-control study, among the 177 samples tested positive for *Campylobacter* spp. significantly higher levels of *Campylobacter* spp. were found in mink kits diagnosed with PADS compared to mink kits without PADS diagnosis.

In the exploratory case study, among the 78 samples that tested positive for MCoV, significantly higher levels of MCoV were found in samples from mink kits diagnosed with PWD compared to mink kits with or without GPD.

**Table 5. The odds ratio (OR) of RT-qPCR findings in fecal samples from mink kits from either a case farm[1] or with a diagnosis of PADS[2] compared to mink kits from a control farm or no PADS diagnosis.**

| Detection of agent by RT-qPCR | Case farm OR[3] (95% CI) | PADS diagnosis OR[3] (95% CI) |
|---|---|---|
| *MD-study* | | |
| MCoV | 0.00 (0.000;3.480) | 0.2708 (0.018;2.583) |
| *Campylobacter* spp. | 7.250 (0.726;107.7) | ∞ (0.533;∞) |
| *Observational Case-control study* | | |
| MCoV | 2.351 (1.027;5.056)* | 1.095 (0.472;2.466) |
| *Campylobacter* spp. | 0.9358 (0.4042;2.288) | 2.060 (0.6505;6.756) |

*95% CI not including one were considered statistically significant.

PADS = Post-weaning Anorexia and Diarrhea Syndrome, MD-study = Method development study.

[1]Definition of case or control farm based on disease history obtained from the consulting veterinarian on the farms.

[2]Diagnosis of PADS based on necropsy findings.

[3]Odds ratio calculated by Fischer's exact test.

An overview of the RT-qPCR and qPCR results in the three studies presented with boxplots with mean, minimum, maximum, and annotations of significance are shown in Fig 1.

## Management data

The management data obtained from the 19 farms did not show any associations between management practice and the diagnosis of PADS on farm level.

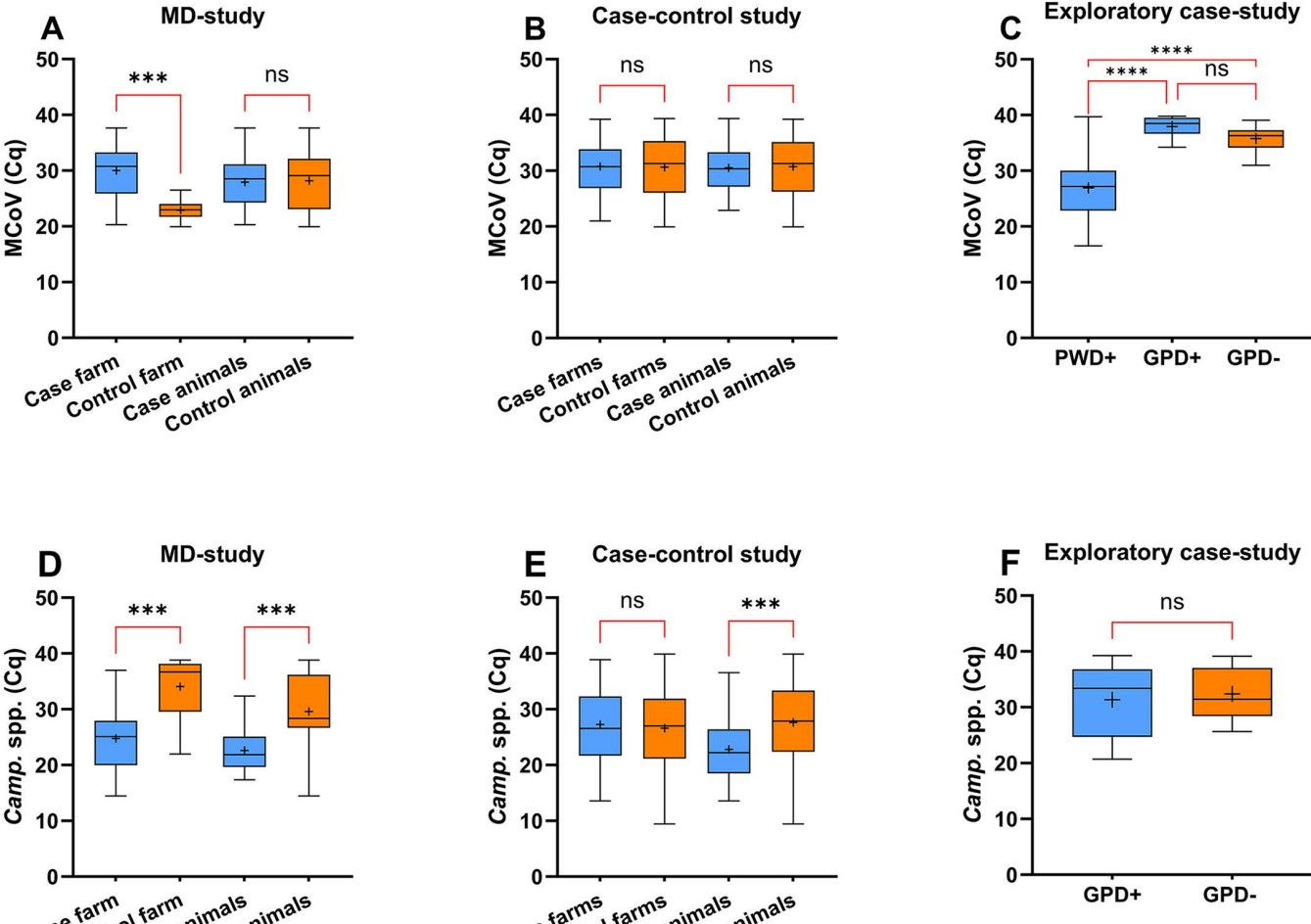

**Fig 1. MCoV RT-qPCR and *Campylobacter* spp. Cq-value distributions in mink kits across three studies** Boxplots illustrating the distribution of Cq-values obtained from RT-qPCR and qPCR analysis of MCoV and *Campylobacter* spp. respectively in fecal samples from mink kits across three studies. **A + D)** Method development study (MD-study) comprising samples from a PADS case farm (*n = 30*) and a control farm (*n = 10*). **B + E)** Observational Study: 208 mink kits (58 cases, 150 controls) from 19 farms. **C + F)** Exploratory Study where 195 mink kits with PWD or GPD analyzed for MCoV and 73 with GPD analyzed for *Campylobacter* spp. Boxplots display median, mean (+), interquartile range (IQR), and minimum/maximum values (whiskers). One-way ANOVA with multiple comparisons was used for statistical analysis with significance level ($p < 0.05$). Significance levels are indicated (***$p < 0.001$, ****$p < 0.0001$, ns: not significant). Case/Control Farm = Farm status based on disease history (obtained by farm veterinarian). Case/Control animals = Diagnosis of Post-weaning anorexia and diarrhea syndrome based on necropsy findings. GPD = Growth period diarrhea. PWD = Pre-weaning diarrhea.

## Next-Generation Sequencing

In total, 75 fecal samples (40 from the MD-study and 35 from the PADS outbreak in 2019) were subjected to Next-Generation sequencing. From those, 17 samples (11 from the MD-study and 6 from the PADS outbreak in 2019) with the highest number of specific reads (> 2000) were selected for further phylogenetic analysis. Of these samples, 14 complete and 3 almost complete MCoV genomes were assembled (GenBank accessions PQ451274-PQ451290, see Table 6 for summary). The sequences of samples 1, 2, 11, 12 and 15 (see Table 6) were obtained by merging 7, 4, 4, 3, and 3 partly overlapping contigs, respectively, while the sequences of the remaining samples could be obtained from single contigs. The reads coverages were in the range of 15–2180, and the values for the multi-contigs sequences represent the length averaged values.

## Phylogenetic analyses

A maximum likelihood tree was constructed based on the 17 novel MCoV genomes and two published MCoV genomes from the Danish MCoV strain MCoV1/11918–1/DK/2015 (MN535737) and the MCoV strain WD1127 from Wisconsin, USA (NC_023760) (Fig 2).

All Danish sequences were *Alphacoronaviruses* and group in a single clade distinct from one strain from the USA. The sequences of the Danish viruses in turn separates into two subclades, where one is constituted by sequences from the MD-study case farm (sample 4–11, see Table 6) and the other by sequences from the 2019 case farms (samples 12–17, see Table 6), the earlier Danish strain from 2015 (MN535737) and the sequences from the MD-study control farm (samples 1–3, see Table 6). Additionally, we can see a further grouping of the four strains from one of the 2019 case farm 1 (samples 12–15, see Table 6), the two strains from the 2019 case farm 2 (samples 16 and 17, see Table 6), and the three strains from the control group together with the 2015 strain.

Table 6. Summary of the 17 MCoV samples selected for assembly and further phylogenetic analysis.

| Sample # | Sample name | Location | Year | Study type | Accession* |
|---|---|---|---|---|---|
| 1 | 45510−5 | Grindsted (DK) | 2018 | MD control | PQ451274.1 |
| 2 | 45510−6 | Grindsted (DK) | 2018 | MD control | PQ451275.1 |
| 3 | 45510−9 | Grindsted (DK) | 2018 | MD control | PQ451276.1 |
| 4 | 45511−22 | Grindsted (DK) | 2018 | MD case | PQ451277.1 |
| 5 | 45511−25 | Grindsted (DK) | 2018 | MD case | PQ451278.1 |
| 6 | 45511−26 | Grindsted (DK) | 2018 | MD case | PQ451279.1 |
| 7 | 45511−27 | Grindsted (DK) | 2018 | MD case | PQ451280.1 |
| 8 | 45511−28 | Grindsted (DK) | 2018 | MD case | PQ451281.1 |
| 9 | 45511−29 | Grindsted (DK) | 2018 | MD case | PQ451282.1 |
| 10 | 45511−32 | Grindsted (DK) | 2018 | MD case | PQ451283.1 |
| 11 | 45511−40 | Grindsted (DK) | 2018 | MD case | PQ451284.1 |
| 12 | 46351−1 | Hjoerring (DK) | 2019 | Outbreak | PQ451285.1 |
| 13 | 46351−3 | Hjoerring (DK) | 2019 | Outbreak | PQ451286.1 |
| 14 | 46351−4 | Hjoerring (DK) | 2019 | Outbreak | PQ451287.1 |
| 15 | 46351−5 | Hjoerring (DK) | 2019 | Outbreak | PQ451288.1 |
| 16 | 46367−3 | Saeby (DK) | 2019 | Outbreak | PQ451289.1 |
| 17 | 46367−5 | Saeby (DK) | 2019 | Outbreak | PQ451290.1 |

DK = Denmark, MD-study = Method development study, specified in section 1.1.1.

*GenBank accession number.

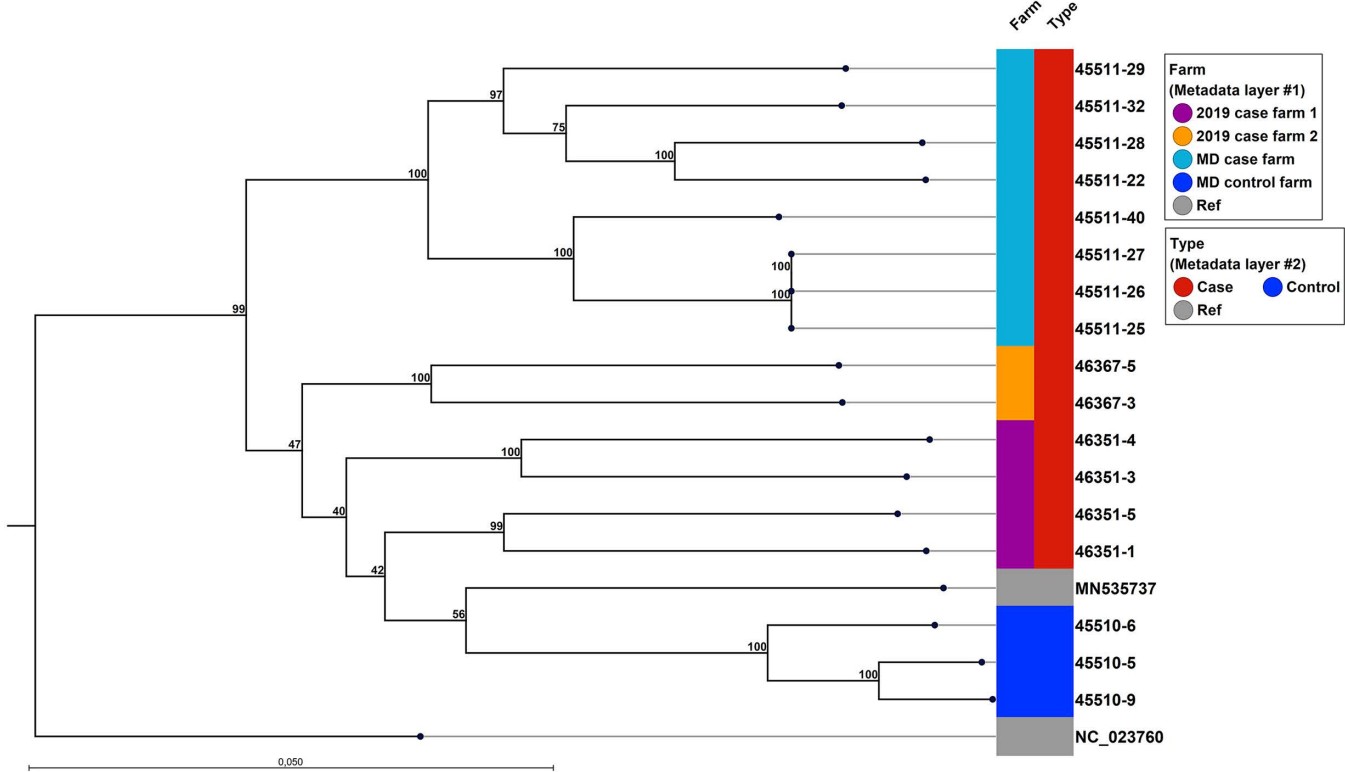

**Fig 2. Maximum likelihood phylogenetic tree of 17 MCoV genomes obtained from Danish farms during 2018−2019 The bootstrap support (100 replicates) are shown at the nodes.** The alignment was cropped prior to phylogenetic calculations leading to sequence lengths in the range of 27832 nt (NC_030292; the outgroup not shown) to 28331 nt (46351−1). The phylogeny comprises 17 near full length MCoV genomes from four Danish farms during 2018−19 (this study) together with a MCoV genome obtained from a Danish mink farm from 2015 (MN535737) and a MCoV strain from Wisconsin, USA sampled in 1998 (NC_023760). The genomes were from Danish farm mink with PADS (type red) and without PADS (type blue). Branch length reflect the degree of divergence of each sequence. The bar shows the length corresponding to 0.05 substitutions per site. Sequences are labelled with sample names. PADS = Post-weaning anorexia and diarrhea syndrome.

## Discussion

To our knowledge, this study represents the first investigation of potential causal factors of PADS in mink kits and provides a tool for large-scale screening for MCoV in fecal samples. A novel RT-qPCR for detection of MCoV was developed. While previous studies have suggested a link between the presence of MCoV and diarrheal syndromes in farmed mink kits [6,7,11], our findings did not support this association in the post-weaning period. However, MCoV was found to be highly prevalent among mink kits on Danish farms throughout the pre-weaning, post-weaning, and growth period, corroborating with serological findings of MCoV antibodies from 1992 in Danish farm mink [8]. Interestingly, higher MCoV levels were observed in fecal samples from kits with PWD compared to those with and without GPD, suggesting a potential decrease in viral levels during the growth period. Further research is required to elucidate the significance of MCoV in relation to PWD and the apparent viral level decline during the growth period.

Phylogenetic analysis showed a strong correlation between phylogenetic grouping and farm of origin, where MCoV genomes isolated from the case farm in the MD-study clustered in their own clade distinct from the MCoV genomes isolated from the control farm. This suggests significant genomic diversity within MCoV, potentially explaining the observed differences in clinical presentation. This finding aligns with the established link between strain variability and pathogenicity in feline coronavirus (FCoV), where distinct biotypes - feline infectious peritonitis virus (FIPV) and feline enteric

coronavirus (FECV) – exhibit different virulence [35]. However, further research is needed to elucidate the potential association between MCoV strain diversity and pathogenicity.

Indications of farm-specific viral strains were observed as variants from the four farms (MD-case, control farm and the two 2019 case-farms) all groups in their own sub-group. In addition, both intra-farm genomic variability and inter-farm phylogenetic clade formation suggest that the specific MCoV strains were maintained for a long time on the farms enabling genomic evolution, and inter-farm transmission events. These are both events that were also observed during the recent SARS-CoV-2 outbreak in farmed mink [36].

The high prevalence of MCoV among Danish farm mink warrants attention due to its potential implications for public health. Seven coronaviruses are known to infect humans, including HCoV-229E, HCoV-NL63 (*Alphacoronaviruses*) and HCoV-OC43, HCoV-HKU1, SARS-CoV, SARS-CoV-2, and MERS-CoV (*Betacoronaviruses*). All known human CoVs are believed to have originated from animals: SARS-CoV-2, SARS-CoV, MERS-CoV, HCoV-NL63, and HCoV-229E likely originated from bats, while HCoV-OC43 and HKU1 are thought to come from rodents (Forni et al., 2017; Su et al., 2016). Recent studies have expanded our understanding of animal-to-human transmission of CoVs, as a novel CoV in a pneumonia patient in Malaysia, classified as canine coronavirus genotype II (CCoV-II) within the *Alphacoronavirus* genus was identified [37]. Subsequently, a nearly identical coronavirus (99.4% genomic similarity) was isolated from a person returning from Haiti [38]. Although bats are considered the primary natural reservoir of *Alpha-* and *Betacoronaviruses* [14], these findings suggest that other host species can act as reservoirs of human CoVs. This underscores the public health threat of animal CoVs and the need for enhanced surveillance. Furthermore, the potential for farm animals to act as mixing vessels for new emerging human CoVs warrants attention [39]. Given the established human-to-mink and mink-to-human transmission of SARS-CoV-2 [11,40], the high prevalence of MCoV in Danish mink raises concerns about their potential role in the emergence of novel human CoVs. Continued research and surveillance are crucial to assess and mitigate this risk.

Additionally, *Campylobacter* spp. were found to be highly prevalent in mink kits during the post-weaning and growth periods, with a potential association between fecal *Campylobacter* spp. levels and PADS diagnosis.

Previous studies have associated *Campylobacter jejuni* as a causative agent in reproductive issues and nursing disease in lactating mink [20,41], and both *C. jejuni* and *C. coli* have been isolated from diarrheal mink kits during weaning [22]. However, isolation of *C. jejuni* in feces from clinically healthy mink at pelting suggests that *C. jejuni* may not be causally linked to gastrointestinal disease at this stage [9]. A limitation in this study is the use of a qPCR assay targeting the 16S rRNA gene, which enables detection at the genus level but lacks species specificity. Further investigation is required to clarify the specific role of *Campylobacter* spp. in mink health, particularly in relation to PADS and potential differences in pathogenicity across life stages. This is particularly relevant given that previous research on Danish wildlife indicated that wild mammals and birds harbor *C. jejuni* strains with serotype distributions and clonal lines that differ significantly from those found in humans and commercial poultry [42]. Possibly mink might also harbor specific strains as part of their natural microbiota and our current data cannot distinguish whether PADS is associated with a specific, pathogenic Campylobacter type or if the higher levels observed in cases simply reflect an opportunistic proliferation of naturally occurring commensal species following intestinal dysbiosis. Future research utilizing species-specific assays or metagenomic typing is necessary to determine if specific Campylobacter strains are unique to PADS cases.

Furthermore, *Campylobacter* spp. is the leading bacterial cause of human gastroenteritis worldwide, with approximately 1 in 10 individuals affected [43]. This genus includes zoonotic pathogens, some of which are highly pathogenic [44]. Human campylobacteriosis is predominantly attributed to *Campylobacter jejuni* and *Campylobacter coli*, although other species within the genus have also been identified in clinical samples [45–47]. Thus, the detection of *Campylobacter* spp. in fecal samples from mink kits during the post-weaning period in this study emphasizes the importance of rigorous hygiene practices on mink farms.

## Conclusion

In conclusion, MCoV was found to be highly prevalent in samples from mink included in this study through the pre-weaning period as well as the post-weaning and growth period. Analyses of case-control samples from farms with and without PADS indicated an association with MCoV strain variant pathogenicity rather than presence or levels of MCoV. This warrants further studies to investigate the impact of MCoV strain variability in relation to diarrhea problems in farm mink. PCR results indicated that *Campylobacter* spp. was highly prevalent in mink during the post-weaning period. It may be relevant to further investigate the role of *Campylobacter* spp. in relation to PADS and other diseases occurring during this period. The result of this study indicates that *Campylobacter* spp. may pose a zoonotic risk for persons handling mink.

## Acknowledgments

We express our gratitude to the breeders who participated in this project, providing valuable assistance and generously donating animal carcasses for the study. We are also grateful for the contributions from veterinary students Mia Berg and Caroline Berner to assist with necropsies.

## Author contributions

**Conceptualization:** Michelle Lauge Quaade, Anne Sofie Vedsted Hammer.

**Data curation:** Michelle Lauge Quaade, Mikael Leijon, Mikhayil Hakhverdyan, Karin Mundbjerg, Anne Sofie Vedsted Hammer.

**Formal analysis:** Michelle Lauge Quaade, Mikael Leijon, Mikhayil Hakhverdyan.

**Funding acquisition:** Anne Sofie Vedsted Hammer.

**Investigation:** Michelle Lauge Quaade, Mikael Leijon, Mikhayil Hakhverdyan, Lars Andresen, Karin Mundbjerg.

**Methodology:** Michelle Lauge Quaade, Mikhayil Hakhverdyan, Lars Andresen, Anne Sofie Vedsted Hammer.

**Project administration:** Michelle Lauge Quaade, Anne Sofie Vedsted Hammer.

**Resources:** Anne Sofie Vedsted Hammer.

**Supervision:** Thomas Bruun Rasmussen, Charlotte Kristiane Hjulsager, Anne Sofie Vedsted Hammer.

**Visualization:** Michelle Lauge Quaade, Mikael Leijon.

**Writing – original draft:** Michelle Lauge Quaade.

**Writing – review & editing:** Mikael Leijon, Mikhayil Hakhverdyan, Thomas Bruun Rasmussen, Charlotte Kristiane Hjulsager, Lars Andresen, Karin Mundbjerg, Anne Sofie Vedsted Hammer.

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
