## [Decision Letter · Decision Letter 0]

4 Nov 2025

Dear Dr. Quaade,

We look forward to receiving your revised manuscript.

Kind regards,

Baochuan Lin, Ph.D.

Academic Editor

PLOS ONE

Journal Requirements:

Reviewers' comments:

Reviewer's Responses to Questions

**Comments to the Author**

1. Is the manuscript technically sound, and do the data support the conclusions?

Reviewer #1: Yes

Reviewer #2: Yes

2. Has the statistical analysis been performed appropriately and rigorously?

Reviewer #1: Yes

Reviewer #2: Yes

3. Have the authors made all data underlying the findings in their manuscript fully available?

Reviewer #1: Yes

Reviewer #2: Yes

4. Is the manuscript presented in an intelligible fashion and written in standard English?

Reviewer #1: Yes

Reviewer #2: Yes

Reviewer #1: The manuscript under review meets all the requirements necessary for publication in the journal. It has been prepared with great care, in accordance with all the rigors of scientific writing. Its advantages include not only a summary of the results of zoonotic safety research conducted to date on Danish American mink farms, but also an indication of further necessary stages of experimental activities in the breeding of the most important species of carnivorous fur animals.

Reviewer #2: This manuscript appears to be a good reflection of sound science to examine disease, post-weaning anorexia

diarrhea syndrome (PADS) in farmed mink.

There are a few areas that need to be clarified: Methods 1.3 Whole Genome Shotgun sequencing is not exactly clear. The procedure is specific for RNA- RNA viruses. In Results section 2.4 this is called Metagenomic shotgun sequencing. Neither description is exactly correct. It is more correctly metagenomic since the filtering of cells and bacteria appears to be a process that was not successful. However, it is also only an examination of isolated RNA sequences. Metagenomes would include the isolation and sequencing of DNA also. Just define this so that readers know you only examined RNA- to find MCoV. I thought there might be information about the Campylobacters based on the metagenomic statement.

Related to Campylobacter, there are many species that may reside naturally in the mink. Others have noted that wild animals may harbor specific types of C. jejuni that differ from those associated with domestic animals. Additionally, there may be other Campylobacter species that are specifically found just in the PADS mink. I believe you should note the limitations of only performing qPCR for 16S rRNA detection.

**Do you want your identity to be public for this peer review?** For information about this choice, including consent withdrawal, please see our Privacy Policy

Reviewer #1: No

Reviewer #2: No

---

## [Author Response · Author response to Decision Letter 1]

17 Dec 2025

Response to the Academic Editor

1. PLOS ONE Style Requirements

Please ensure that your manuscript meets PLOS ONE's style requirements...

Response: We have carefully reviewed the PLOS ONE style requirements. We confirm that the manuscript, including file naming conventions and formatting, has been updated to fully adhere to these guidelines.

2. Data Availability

Please confirm at this time whether or not your submission contains all raw data required to replicate the results of your study...

Response: We have revised the Data Availability Statement to ensure full transparency. The "minimal data set" required to replicate all study findings, including necessary metadata, has been deposited in a stable, public repository.

• Revised Data Availability Statement: Data from this study is available at the Open Science Framework (OSF) under study ID m4ce2 and can be accessed via DOI: 10.17605/OSF.IO/M4CE2. All MCoV sequence data has been deposited at NCBI with accessions: PQ451274.1-PQ451290.1.

3. Additional Citations

If the reviewer comments include a recommendation to cite specific previously published works...

Response: The reviewers did not recommend specific citations in their initial comments. However, to address Reviewer #2’s comment regarding Campylobacter diversity, we have proactively added a relevant reference (Petersen et al., 2001) to support our discussion on wildlife-specific strains.

4. Reference List

Please review your reference list to ensure that it is complete and correct.

Response: We have thoroughly reviewed the reference list and confirmed it is complete, correct, and compliant with journal formatting.

Response to Reviewer #2

Comment 1: Sequencing Terminology

Methods 1.3 Whole Genome Shotgun sequencing is not exactly clear... In Results section 2.4 this is called Metagenomic shotgun sequencing. Neither description is exactly correct... Just define this so that readers know you only examined RNA.

Response: We agree with the reviewer that the previous terminology was imprecise given that our protocol specifically targeted RNA. To improve clarity and accuracy we have renamed the headers for sections 1.3 and 2.4 to "Next-Generation Sequencing."

Comment 2: Campylobacter Specificity and Wildlife Strains

Related to Campylobacter, there are many species that may reside naturally in the mink. Others have noted that wild animals may harbor specific types of C. jejuni that differ from those associated with domestic animals... I believe you should note the limitations of only performing qPCR for 16S rRNA detection.

Response: We thank the reviewer for this important observation regarding Campylobacter diversity. We agree that a significant limitation of our study is the reliance on a 16S rRNA-based qPCR, which detects the genus but does not distinguish between species. We acknowledge that mink likely harbor natural/commensal Campylobacter species and that specific types—potentially different from those in other domestic animals—might be the true drivers of PADS. We have updated the Discussion (Page 25, Lines 460-470) section to explicitly address this limitation, noting that we could not differentiate between a specific pathogenic species unique to PADS and a general overgrowth of commensal flora.

Reviewers comments:

Reviewer #2: This manuscript appears to be a good reflection of sound science to examine disease, post-weaning anorexia diarrhea syndrome (PADS) in farmed mink. There are a few areas that need to be clarified: Methods 1.3 Whole Genome Shotgun sequencing is not exactly clear. The procedure is specific for RNA- RNA viruses. In Results section 2.4 this is called Metagenomic shotgun sequencing. Neither description is exactly correct. It is more correctly metagenomic since the filtering of cells and bacteria appears to be a process that was not successful. However, it is also only an examination of isolated RNA sequences. Metagenomes would include the isolation and sequencing of DNA also. Just define this so that readers know you only examined RNA- to find MCoV. I thought there might be information about the Campylobacters based on the metagenomic statement.

Response: The titles of paragraphs 1.3 and 2.4 has been changed to “Next-Generation Sequencing” to improve clarity.

Related to Campylobacter, there are many species that may reside naturally in the mink. Others have noted that wild animals may harbor specific types of C. jejuni that differ from those associated with domestic animals. Additionally, there may be other Campylobacter species that are specifically found just in the PADS mink. I believe you should note the limitations of only performing qPCR for 16S rRNA detection.

Response: We thank the reviewer for this insightful observation regarding the specificity of our assay and the diversity of Campylobacter in wildlife.

We fully agree that a significant limitation of our study is the reliance on a 16S rRNA-based qPCR, which detects the genus as a whole but does not distinguish between specific species or strains. We acknowledge that mink likely harbor natural, commensal Campylobacter populations, and as the reviewer notes, studies indicate that wildlife often carry specific strains or clonal lines that differ from those found in humans or other domestic animals.

To address this, we have revised the Discussion section (Page 25, Lines 460-470). We now explicitly state that our data cannot distinguish between specific pathogenic types and commensal flora. We have also incorporated a reference to Petersen et al. (2001) to highlight that Campylobacter strains in Danish wildlife have been shown to possess distinct serotype distributions and clonal lines compared to agricultural animals, suggesting that the "wildlife reservoir" may be distinct from the pathogenic strains usually screened for.

---

## [Decision Letter · Decision Letter 1]

5 Jan 2026

Association between mink coronavirus (MCoV), Campylobacter spp., and diarrhea in farmed mink (Neogale vison)

PONE-D-25-37216R1

Dear Dr. Quaade,

We’re pleased to inform you that your manuscript has been judged scientifically suitable for publication and will be formally accepted for publication once it meets all outstanding technical requirements.

Kind regards,

Baochuan Lin, Ph.D.

Academic Editor

PLOS One

Additional Editor Comments (optional):

Reviewers' comments:

Reviewer's Responses to Questions

**Comments to the Author**

Reviewer #2: All comments have been addressed

2. Is the manuscript technically sound, and do the data support the conclusions?

Reviewer #2: Yes

3. Has the statistical analysis been performed appropriately and rigorously?

Reviewer #2: N/A

4. Have the authors made all data underlying the findings in their manuscript fully available?

Reviewer #2: Yes

5. Is the manuscript presented in an intelligible fashion and written in standard English?

Reviewer #2: Yes

Reviewer #2: The authors have written a nice paper and have addressed the issues that I identified in the original submission.

**Do you want your identity to be public for this peer review?** For information about this choice, including consent withdrawal, please see our Privacy Policy

Reviewer #2: No

---

## [Editor Report · Acceptance letter]

PONE-D-25-37216R1

PLOS One

Dear Dr. Quaade,

I'm pleased to inform you that your manuscript has been deemed suitable for publication in PLOS One. Congratulations! Your manuscript is now being handed over to our production team.

Kind regards,

on behalf of

Dr. Baochuan Lin

Academic Editor

PLOS One